# Non-Viral Delivery of RNA Gene Therapy to the Central Nervous System

**DOI:** 10.3390/pharmaceutics14010165

**Published:** 2022-01-11

**Authors:** Ellen S. Hauck, James G. Hecker

**Affiliations:** 1Department of Anesthesiology, Lewis Katz School of Medicine, Temple University, 3401 N Broad St., Philadelphia, PA 19140, USA; 2Department of Anesthesiology, Harborview Medical Center, University of Washington, P.O. Box 359724, 329 Ninth Ave, Seattle, WA 98104, USA; heckerj@uw.edu

**Keywords:** non-viral, lipid-mediated, gene delivery, transfection, RNA, DNA, siRNA, neurons, post-mitotic, molecular therapy, transient, CHO, NIH3T3

## Abstract

Appropriate gene delivery systems are essential for successful gene therapy in clinical medicine. Lipid-mediated nucleic acid delivery is an alternative to viral vector-mediated gene delivery and has the following advantages. Lipid-mediated delivery of DNA or mRNA is usually more rapid than viral-mediated delivery, offers a larger payload, and has a nearly zero risk of incorporation. Lipid-mediated delivery of DNA or RNA is therefore preferable to viral DNA delivery in those clinical applications that do not require long-term expression for chronic conditions. Delivery of RNA may be preferable to non-viral DNA delivery in some clinical applications, since transit across the nuclear membrane is not necessary, and onset of expression with RNA is therefore even faster than with DNA, although both are faster than most viral vectors. Delivery of RNA to target organ(s) has previously been challenging due to RNA’s rapid degradation in biological systems, but cationic lipids complexed with RNA, as well as lipid nanoparticles (LNPs), have allowed for delivery and expression of the complexed RNA both in vitro and in vivo. This review will focus on the non-viral lipid-mediated delivery of RNAs, including mRNA, siRNA, shRNA, and microRNA, to the central nervous system (CNS), an organ with at least two unique challenges. The CNS contains a large number of slowly dividing or non-dividing cell types and is protected by the blood brain barrier (BBB). In non-dividing cells, RNA-lipid complexes demonstrated increased transfection efficiency relative to DNA transfection. The efficiency, timing of the onset, and duration of expression after transfection may determine which nucleic acid is best for which proposed therapy. Expression can be seen as soon as 1 h after RNA delivery, but duration of expression has been limited to 5–7 h. In contrast, transfection with a DNA lipoplex demonstrates protein expression within 5 h and lasts as long as several weeks after transfection.

## 1. Introduction

Gene therapy has the potential to significantly advance clinical medicine, but the risks and duration of gene delivery should be closely matched to the proposed clinical application [1]. Long-term expression after gene therapy is useful for diseases that require chronic levels of protein expression, such as inherited enzyme deficiencies or cancer, and for these diseases viral vectors may offer advantages. For clinical applications in which only short-term gene expression is required or warranted, the delivery of nucleic acids (DNA or RNA, including mRNA, siRNA, shRNA, and microRNAs saRNA) by means of non-viral vectors, particularly chemical-based carriers, such as cationic lipids, cationic polymers, or lipid nanoparticles, provide a far more favorable risk/benefit ratio. Examples might include pre-operative neuro- or other organ protection, rapid expression of therapeutics after stroke, spinal cord injury or traumatic brain injury, CNS expression of replacement, or interference oligonucleotides for any disease with a known cDNA target sequence, among many others. Lipid/polymer-mediated transfection offers other advantages over viral vectors, most notably safety, low immunogenicity, and the ability to deliver payloads of nearly unlimited size [2,3]. Figure 1 has schematic diagrams of both viral vector and non-viral vector transfection. Figure 1A shows the transfection pathways taken by lentiviruses and adenoviruses, two of the common viruses used for gene therapy. In both cases the desired gene DNA makes its way into the nucleus and may integrate into the host genome. Figure 1A shows the transfection pathways taken by non-viral vectors. In this figure, the vectors again carry DNA, which enters the nucleus. Here the DNA is unlikely to integrate into the host genome. Non-viral vectors may also package different types of RNAs, many of which do not need to enter the nucleus. Transfection with RNA was initially assumed to be useful for short-term expression of a therapeutic, but interest in RNA delivery, specifically mRNA, but also including siRNA, shRNA, and miRNA, as long-term treatment for multiple diseases has not only gained traction, but became a reality in the form of COVID-19 vaccines [4].

Cationic lipid- or polymer-mediated gene transfer is particularly suited for transient gene expression, both in basic research and in selected clinical applications. Cationic lipids are commonly comprised of a polar headgroup and non-polar symmetric or dissymmetric carbon-based tail and may also include nuclear localization signals (NLS), antibodies, polymers, or other targeting moieties. Negatively charged nucleic acids condense and self-assemble into heterogeneous complexes when mixed with cationic lipids [6]. The structure and size of these complexes affect transfection efficacy and vary with temperature, concentration, charge ratio, buffer, time, and lipid composition. These lipid/nucleic acid complexes protect the nucleic acids from degradation in the extracellular environment [7,8]. Numerous laboratories [9,10,11], including our own [9], have investigated the limiting parameters of cationic lipid-mediated transfection with the goal of improving transfection efficiency. The first cationic polymers used to package nucleic acids were Poly-l-lysine (PLL) and polyethylenimine (PEI). These polymers tend to complex with nucleic acids into smaller and more uniform particles, enhancing transfection efficiency. However, surface charges on both types of complexes have caused some cytotoxicity [10]. Extensive work has also been put into formulating cationic polymers to increase transfection efficiency and decrease toxicity [12,13].

Barriers to lipid or polymer-mediated nucleic acid transfection include: (1) transport of the nucleic acid/lipid complex through the extracellular environment to target tissue(s); (2) association and uptake of the nucleic acid/lipid complex by the target cell [14]; and (3) intracellular nucleic acid release from the nucleic acid/lipid complex [12]. DNA transfection has the added barrier of transport into the cell nucleus. Additional barriers exist for some target tissues, including the central nervous system (CNS), protected by the BBB and largely consisting of non-dividing cells. We present here a review of the progress made in delivery of nucleic acids, specifically RNAs, to the CNS. We will show that many barriers have been overcome.

## 2. Non-Viral Instead of Viral Gene Delivery

Currently, more than 70% of clinical trials investigating nucleic acids as therapy use some form of viral vector, including retroviruses, lentiviruses, adenoviruses, and adeno-associated viruses [13]. These vectors, often at high titer, can provide delivery of the nucleic acid into cell cytoplasm and nuclei, where DNA becomes integrated into the genome, but are limited by the size of the nucleic acid that can be packaged. Integration into the genome is a desirable effect for long-term therapy for illnesses caused, for example, by genetic enzyme deficiencies, such as severe combined immunodeficiency and glycogen storage diseases [15,16]. Early clinical trials with these vectors uncovered problems, such as immunologic (including allergic) reactions [17], off target effects, and indiscriminate integration into the genome, leading to cancers such as leukemia [18]. Since this earlier work, viral vector systems have been modified to increase safety, but recently, another adverse event was reported after the delivery of a viral vector. A patient participating in a phase II trial of AAV.7m8-aflibercept, a viral vector for intravitreal injection to treat diabetic macular edema, lost their eyesight in the treated eye [19]. In addition to serious adverse events, viral vector delivery may not in all cases be a permanent solution to a genetic disease. Heller et al. followed mice with a murine model of Krabbe disease (Twitcher mice) and treated with AAV9-galacytosylceramide gene as neonates. The vector was delivered via one of three methods: intracerebral injection, intrathecal injection, or intravenous injection. The mice demonstrated decreased central and peripheral neuropathology and prolonged survival, but at ages of 6–8 months began to show small focal demyelinating regions in the brain. This suggested dysregulation of therapeutic GALC, likely due to exhaustion of the tranduced episomal DNA [20]. Work will continue on systems using viral vectors to improve their safety in light of the potential benefit of a single curative treatment.

The potential hazards of viral vectors have led many investigators to consider the use of non-viral vectors for delivery of therapeutic nucleic acids, including for treatment of genetic diseases. Patisiran, a non-viral vector in the form of a lipid nanoparticle carrying an siRNA targeted to inhibit transthyretin protein for the treatment of hereditary transthyretin amyloidosis, recently received FDA approval [21]. Patients treated with Patisiran undergo intravenous injection of the vector once every 3 weeks. Nucleic acid delivered to cells by non-viral vectors generally do not integrate into the host genome, although this can be considered as a benefit and a detriment depending on the purpose of the proposed nucleic acid therapy. Historically, nonviral vector delivery efficiency has been lower than that of viral vectors, and a great deal of work has focused on improving both in vitro, ex vivo, and in vivo transfection efficiency, including from our lab [22,23,24,25]. Generally, non-viral vectors have been delivered to cells or tissues of interest by physical or chemical methods. Physical methods include electroporation, sonoporation, laser irradiation, magnetofection, and microinjection [26], which use electricity, sound, lasers, magnetic particles, and micro catheters, respectively, to open holes in cell membranes to allow for entry of foreign nucleic acids. All of these carry the risk of permanent damage to the cells and cell death [27]. Microinjection is the direct injection of nucleic acids into cells and is a technique that requires specially trained personnel at a minimum. All the above methods can transfect a small number of cells, making them useful primarily for ex-vivo cell transfection, followed by injection of the transfected cells into cells or tissues of an organ/organism. Due to the limited clinical usefulness of these techniques, a greater focus has been placed on chemical methods of nucleic acid transfection, more specifically lipid- or polymer-mediated transfection. Advances in chemical non-viral transfection have focused on chemical modifications, such as lipid asymmetry and amino lipids [28,29]. A variety of cationic polymers have been studied as well [24]. These efforts have been directed at improvements in packaging, stability, endosomal escape, cargo release, and targeting of specific tissues or organs. Targeting to tissues and organelles is generally dependent on peptides, antibodies and other proteins. Cargo release of the nucleic acid from the lipid or polymer carrier is dependent on disulfide linkages [30]. Table 1 provides a summary of some of the advances now found in the packaging and stability of nucleic acid delivery vehicles, and see Figure 2 below.

Cationic lipids and polymers have different advantages. Cationic polymers tend to complex with nucleic acids into smaller and more uniform nanoparticles, enhancing transfection efficiency. Their high cationic charge also favors efficient endosomal uptake, but this charge also carries some cytotoxicity [10]. In addition, a full understanding of the role components of these polymers play in transfection makes rational design more elusive and makes translation to clinical applications more challenging [33,34,35]. Cationic lipids have also shown some cytotoxicity and have required extensive formulation work to optimize ideal combinations and concentrations of lipid components. Cationic lipid complexes are not as stable as cationic polymers but newer work has shown promise in this area [36,37,38]. Scale up for cationic lipids is easier [37,39,40] than for cationic polymers, and generally they have shown promise in translating to clinical use, discussed below.

## 3. DNA and RNA as Genetic Therapies

Both DNA and RNA, recognized by the immune system as foreign genetic material, are rapidly degraded in biologic systems. Both nucleic acids therefore present challenges when considered as therapeutics and encapsulating into a viral or non-viral vector is critical. The use of DNA (mostly as cDNA) has been seen as preferable for therapy, requiring long-term expression, such as replacement therapy for genetic diseases. Incorporation of DNA into the genome using non-viral vector delivery remains a theoretical if extremely remote possibility, and even this rare chance is avoided with mRNAs, which are transcribed in the cytoplasm and do not need to reach the nucleus. This is not true of some small RNAs that target gene promoters or for CRISPR applications. Delivery of DNA has the additional barriers of transport through the cytoplasm, avoiding degradative pathways and transit across the nuclear membrane. Episomal DNA residing in the cytoplasm is likely not permanent, for a variety of reasons [37,41]. Episomal gene DNA can be lost by recombination, by destruction with nucleases, and by partitioning into nonnuclear cell compartments. The DNA can also be lost through cell division or can experience loss of function by silencing of perceived foreign DNA.

Although there are numerous successes in vitro and in animal models using both DNA and RNA, RNA has the benefit of being more efficiently transfected into non-dividing cells [42]. One of the mechanisms for foreign DNA entrance into the nucleus is through cell division. Although somewhat less stable than DNA, RNA synthesized in vitro can be modified to improve its stability and decrease immunogenicity. Many RNAs of smaller size (siRNA, saRNA, shRNA, miRNA) are easier to protect and package and, using them, it is possibly easier to achieve efficient transfection. Some of these small RNAs must enter the nucleus to show effect, and greater translation efficiency may translate to greater numbers of molecules reaching this destination. mRNA, once in the cell, is rapidly translated, does not enter the nucleus, and cannot integrate, so may be best for short-term therapy. Increased transfection efficiency in non-dividing cells, such as neurons, makes RNA the preferred nucleic acid for transfection of the CNS [42]. For all these reasons, RNA appears to be optimal for repetitive CNS gene therapies. Even for genetic disorders, the risk and benefits of viral vectors, their expression levels and the duration of expression needed must be weighed against the advantages of RNA and DNA in specific disease applications.

## 4. Cationic Lipids and Lipid Nanoparticles

Cationic lipids have been used to protect nucleic acids and increase the efficiency in the process of transfection for decades. Felgner et al defined three functional properties needed for transfection using cationic lipids: spontaneous capture of the nucleic acid by the cationic lipid to form lipoplexes, increased cellular uptake, due to interaction of the positively charged complexes with biological surfaces, and fusion with the plasmalemma or endosome [43]. The lipoplexes formed in early experiments increased transfection efficiency but also showed some cytotoxicity manifesting both as apoptosis and inhibition of the immune system [44]. Much work has been done in recent years to improve efficiency and decrease the toxicity of these lipoplexes. All aspects of the structure of these cationic lipids have been examined; toxicity has been linked to the cationic head group and its multivalent nature, the length and degree of unsaturation of the hydrophilic domain chains, and the linking functionalities. Efficiency of transfection is related to all these structural features, and variations of each can improve transfection efficiency for the target cell or organ [45,46].

Lipid nanoparticles are another means to carry nucleic acid into cells/tissues/organisms and have been used in the pharmaceutical industry for the delivery of poorly soluble drugs. Their safety is well known for dermal, ocular, and oral administration, but the use of surfactants for stabilization may induce inflammatory reactions in other routes of administration [29]. Investigation of their use for carrying genetic material became more feasible with the use of lipoplexes, allowing the encapsulation of a more neutral drug. Composition of these nanoparticles can vary widely, but, as an example, the lipid nanoparticles created to carry the mRNA for the SARS-CoV2 vaccines are composed of the cationic lipid:mRNA complex, an added polyethylglycosylated lipid for stability, as well as a phospholipid and cholesterol for structure [33]. As with the simpler lipoplexes (cationic lipid: nucleic acid complexes), the specific optimization of these particles can improve transfection efficiency and target the particles to tissues/organs [3]. Due to the more complex nature of lipid nanoparticles compared to lipoplexes, their synthesis is far more involved than creating cationic lipid:nucleic acid complexes and can involve the use of high shear homogenization, ultrasound treatments, solvent emulsification, and evaporation, although these processes are well known in industry [34]. Recently, reports of simplified chemical structures, as well as simplified synthesis for LNP:RNA complexes, have been reported. Kuboyama et al. describe a cationic lipid (3-hydroxylpropyl)dilinoleyamine) that can be made in a one step process. This lipid can then be mixed with ApoB-siRNA and an ethanolic solution of lipids, including this cationic lipid above, then filtered, diluted, and delivered to culture cells and/or mice. Measurement of ApoB levels showed that inhibition of ApoB synthesis occurred both in vitro and in vivo and that no adverse side effects were observed in the mice [35]. Blakney et al. reported that saRNA can be complexed with formulated LNPs such that the nucleic acid is located on the exterior of the nanoparticle. These LNP:saRNA complexes were shown to deliver saRNA-fluc with the same efficiency as LNPs encapsulating the saRNA, both in cultured cells and in BALB/c mice [38], suggesting that saRNA is as well protected from degradation when adhering to the external surface of the LNP as when encapsulated. This might suggest the possibility of a useful “generic” LNP. Work continues increasing stability, decreasing toxicity, ease of synthesis, and efficiency of transfection for both cationic lipids and LNPs. To date, the only genetic drug on the market using a non-viral delivery system, Patisiran, is encapsulated in a lipid nanoparticle [21].

## 5. DNA/RNA Transfection into the Central Nervous System

Central nervous system diseases, such as Parkinson’s disease, Alzheimer’s disease, and brain tumors, have been resistant to traditional therapies, at least in part due to the nearly impenetrable blood brain barrier (BBB). Gene therapy represents a new avenue for treating these diseases and a great deal of research has been focused on such approaches. Several studies have shown therapeutic benefits of cationic lipid: siRNA or LNP:siRNA transfection in the reduction of glioblastoma cells either in vitro or when tumor cells have been seeded subcutaneously in mice (ex vivo), but the BBB is not a factor in these studies +++ [47]. Delivery of pDNA remains the focus of most studies; in a recent review, Wang et al. listed 20 papers discussing nucleic acid delivery to the CNS, but only 6 described transfection of RNA [48]. Delivery of nucleic acids as medications/therapy to the CNS faces all the challenges of delivery to any other organ of the body, such as rapid biodegradation of nucleic acids and immunogenicity of nucleic acid breakdown products, but in addition, delivery to the CNS must face the hurdle of the blood brain barrier (BBB). There are two approaches to the blood brain barrier. The first approach is avoidance of the BBB. This includes intrathecal injections, intraparenchymal injections, and breakdown of the BBB by creating a localized inflammatory reaction causing a transient opening of tight junctions with, for example, mannitol, or via an event such as traumatic brain injury [49,50]. The alternate approach is to adapt the drug to use known BBB transport systems [51]. Both methods have experienced successes [52].

Intracranial injection of nucleic acids complexes was one of the first approaches to bypassing the BBB. Intracortical injection in mice of pVEGF-GFP complexed with cationic lipid showed angiogenesis in the regions surrounding the injection/transfection sites [53]. This approach has also been used for the transfection of RNAi. Pfeifer et al. were able to inject RNAi designed to suppress prion protein expression in scrapie-infected mice. Here, the RNAi was encapsulated into a lentiviral vector. The results were similar in that results were positive but only locally around the site of the intraparenchymal injection [54].

Transfection of the CNS with nucleic acids in lipoplexes can also be achieved by delivery of the lipoplex at a time when the BBB is disrupted. Intranasal delivery of a cationic lipid: GFP-mRNA was successful [55] in demonstrating short-term expression of GFP when 3 mg/kg (a high dose) of the complex was given to mice intranasally. Dosing for intranasal delivery has been suggested to require such a high volume of drug that it disrupts the BBB at least temporarily [52]. Nasal delivery of the therapeutics has been demonstrated in mouse models and likely works by transient disruption of the BBB at a high dose [50,52].Delivery of siRNA:LNP to the CNS has also been shown after intravenous injection in mice having undergone traumatic brain injury [56].

Another avenue for crossing the BBB is to use the endogenous transport mechanisms within the BBB. For small molecule delivery, there are carrier-mediated transporters (CMT). These transporters take vitamins and other nutrients into the brain. CMT systems include, for example, GLUT1, LAT1 (large neutral amino acid transporter), and CAT1 (cationic amino acid transporter). LAT1 is the transport system that brings L-DOPA, gabapentin, and AAV9 into the brain [52]. Other molecules rely on receptor-mediated transporters to gain access to the CNS. The BBB insulin receptor and the transferrin receptor are examples. Taking advantage of the AAV9 vector’s transportation through the BBB, 15 pediatric patients with spinal muscular atrophy 1 were given one intravenous dose of an AAV9 vector carrying a replacement cDNA-survival motor neuron 1 gene. Two years after treatment, all 15 patients were alive and event free compared to the 8% survival of an historical cohort [57]. Receptor-mediated transporters on the BBB have also been explored as mechanisms to cross the BBB. These transporters must be targeted with transporter specific ligands. Zhang et al. synthesized “immunoliposomes” encapsulating cDNA-LacZ. This delivery system carries a monoclonal antibody to the human insulin receptor, allowing the immunoliposome to cross the BBB. When injected intravenously into the rhesus monkey, beta-galactosidase was found to be widely expressed throughout the CNS [56]. These delivery systems have been named Trojan horse liposomes (THLs) [52]. Delivery of siRNA has also been reported. Wang et al. created a lipid nanoparticle that carried both the GCN peptide for receptor-mediated transport across the BBB and the tet1 peptide to target receptors on neurons in the CNS. This LNP encapsulated an siRNA to β-site amyloid precursor protein cleaving enzyme 1 (BACE 1). Reduction of BACE 1 activity would decrease the production of the amyloid precursor involved in the development of AD. Double transgenic (APP/PSI) mice (model system for AD) were given an intravenous injection of this LNP:siRNA-BACE 1 and sacrificed 1 h later. Immunohistochemistry demonstrated a 50% reduction in BACE 1 mRNA, as well as a reduction in amyloid plaques [58].

The BBB can be bypassed by direct injection. Thakker et al. used intrathecal injection of an siRNA to demonstrate widespread inhibition of protein expression [59]. In this case, naked siRNA targeted for sequences coding for enhanced green fluorescent protein (EGFP) was injected into the dorsal third ventricle of mice overexpressing EGFP. Decreased expression of the protein was seen in brain tissue, particularly close to the site of injection. To demonstrate more widespread effectiveness, siRNA targeted to inhibit expression of the dopamine transporter protein (DAT-1) was similarly injected as naked siRNA into the dorsal third ventricle. A significant decrease in DAT-1 was seen throughout brain tissue. Intracerebroventricular injection into mice has also been used to deliver a non-viral vector carrying siRNA into Thy1-aSyn mice, a model system for Parkinson’s disease. Here, siRNA against α-synuclein (SNCA) was complexed with a branched PEI F25-LMW and injected into the lateral ventricle of the mice. Five days after transfection, mice were sacrificed and SNCA mRNA expression in the striatum was found to be reduced by 65%, and SNCA protein by 59%, compared to control animals [60].

In our laboratory, we have demonstrated repeatedly in a variety of species and over many years the rapid and widespread distribution and expression of a variety of reporter and neuroprotective gene sequences by intrathecal injection of the nucleic acid complex. Anderson et al. demonstrated the stability of cationic lipid:mRNA complexes in both human and rat cerebral spinal fluid (CSF) [61]. The cationic lipid complex preserved mRNA for at least 4 h compared to less than 5 min for naked mRNA. We have also shown that intrathecal delivery via the cisterna magna and lateral ventricle of cationic lipid:pDNA complexes produces widespread protein expression, as well as biological effect. Transfection by delivery to the cisterna magna of rats of a cationic lipid:pDNA-fLUC complex provides widespread CNS expression of luciferase, which peaks at approximately 72 h but is visible by real time bioluminescence imaging using the In Vivo Imaging System (IVIS) by Caliper Life Sciences (Hopkinton, MA, USA) for at least 10 days after transfection [62]. In addition, in a mouse model, a pDNA coding for an immunomodulatory fusion protein, OX40-TRAIL, was complexed with MLRI and delivered into the cisterna magna of mice prior to the onset of experimental autoimmune encephalomyelitis (EAE), a model for multiple sclerosis (MS). Transfected mice showed decreased severity of disease [23]. Using the same cationic lipid, myristoyl lauroyl Rosenthal inhibitor (MLRI), complexed to mRNA-fLUC translated in vitro from the pDNA-fLUC construct and delivered by cisterna magna injection into rats, we found luciferase expression, again by IVIS imaging. Rats were dosed with 150 μg/kg mRNA, complexed with MLRI by cisterna magna injection of a total of 120 μL delivered over 15 min. Within 1 hour after transfection, rats were imaged repeatedly every hour to look for luciferase expression. At each time point, animals were injected with an intravenous dose of luciferin and immediately imaged. Imaging was repeated every 2 min until light was no longer detected. The purpose of this repeated imaging was both to detect the most intense period of light emission from the rat after a single dose of luciferin and to confirm that light was no longer detectable before the next hourly time point and luciferin dose. Figure 3 shows a schematic diagram of the experiment just described.

Figure 4 shows images from one such minute-to-minute time course. As is shown in the figure, peak light emission occurs very rapidly (1–3 min) and light is no longer detected 15 min after intravenous injection of luciferin. To report only light due to luciferin breakdown, background light counts were taken prior to each luciferin injection and subtracted from counts measured after luciferin injection. Light emission was detected up to 7 h after transfection. Figure 5 shows a representative hour-to-hour time course, demonstrating luciferase expression over hours after transfection. Again, background counts are subtracted from total counts measured at each time point. To date, these remain the only in vivo images that demonstrate transfected mRNA functioning after CSF injection in a live animal. Finally, one rat was sacrificed 3 h after transfection with the MLRI:mRNA-fLuc lipoplex (via the cisterna magna). Brain tissue from this transfected animal, as well as from a control animal, were sectioned and processed for immunohistochemistry, as previously described [62] erase was detected using a rabbit monoclonal antibody against luciferase. Figure 6 shows several sections of brain tissue at different magnifications. Luciferase was widely detected in the brain tissue of the transfected rat, but not in the control rat.

Detection of light by IVIS imaging from luciferin breakdown in the transfected rats required optimization of several experimental parameters. Table 2 summarizes these efforts. In vitro transcription (IVT) efficiency of the pDNA:fLuc plasmid was improved by using the ARCA cap (Thermo-Fisher Scientific, Waltham, MA, USA), and then all IVT mRNA was tested for biologic activity in vitro by transfection into CHO-K1 tissue culture cells. In addition, background light was minimized by screening the rats by IVIS imaging immediately before transfection, as well as shaving the fur off the head of each transfected animal.

## 6. Conclusions

Gene therapy has become an active research avenue for many diseases of the CNS. Ongoing research programs, using model systems for AD, PD, MS, stroke, and brain tumors, are actively searching for cures and treatments of these and other CNS diseases. Once candidate treatments are found, delivery must contend with the BBB and transfection into a largely nondividing cell population. Both viral and non-viral vectors have been used to deliver gene therapy to the CNS, but non-viral vectors have several advantages, including lower toxicity, lower immunogenicity, and the ability to package a large nucleic acid load. Non-viral vectors include physical transfection methods, but lipid-based non-viral vectors, such as cationic lipid, nucleic acid complexes, and lipid nanoparticles encasing nucleic acid, have greater potential for whole organ transfection or even full systemic transfection and have been a greater focus of research efforts. Decades ago, cationic lipids were first complexed with nucleic acids to protect them and also to neutralize the strongly negative charge of these macromolecules. Since that time, chemical manipulation of cationic lipids, as well as their incorporation as part of LNPs, have increased transfection efficiency and decreased toxicity and immunogenicity. More recently, there has been an increased interest in the use of RNA for genetic therapy to the CNS. This is at least in part due to the fact that transfection of nondividing cells with RNA is more efficient than transfection with DNA. While RNA-based gene therapy is shorter lived than DNA therapy, we have shown effects lasting longer than might be expected. Although there have been only a few examples of FDA approved RNA drugs, Patisiran is an siRNA treatment for a genetic disease that is given intravenously once every 3 weeks, a reasonable treatment regimen.

Intrathecal injection is the means we have used to successfully transit the BBB and transfect the CNS with single CSF injections. We have shown that transfection with both pDNA and mRNA is possible, efficient, and can be followed by a time course using a reporter gene and in vivo imaging. There has been tremendous progress in overcoming the road blocks to gene therapy for the CNS, and a focus on RNA-based therapies, as well as delivery methods that cross the BBB, will likely increase the speed of this progress in the years to come.

## Figures and Tables

**Figure 1 pharmaceutics-14-00165-f001:**
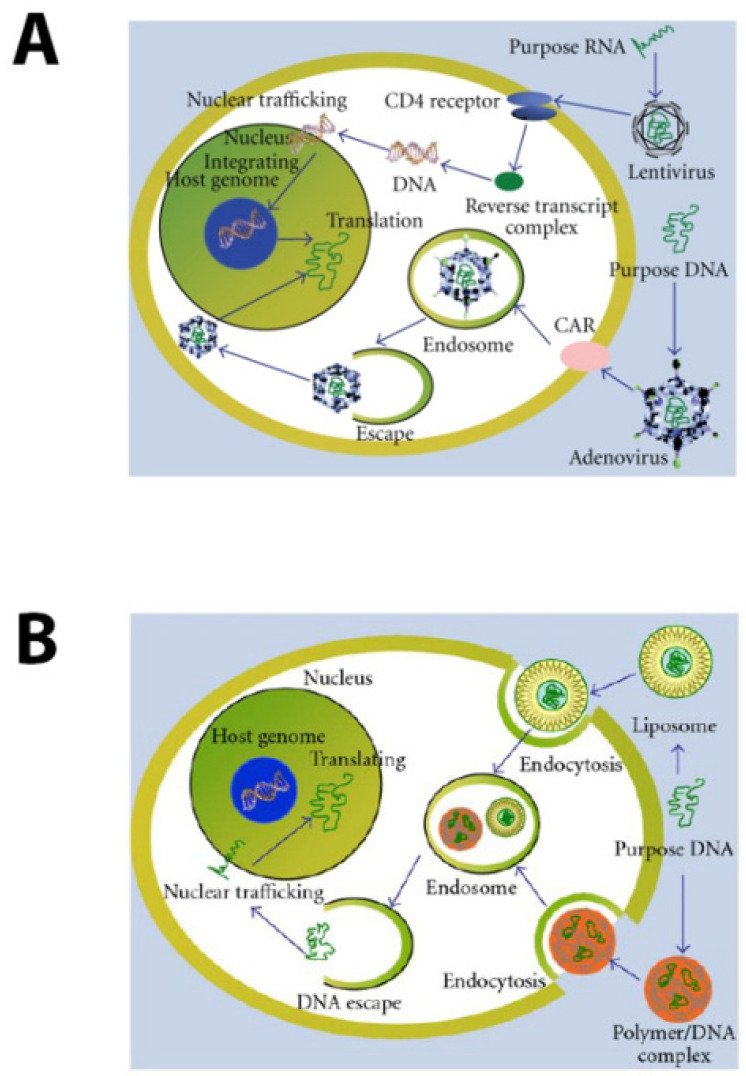
Schematic diagrams of viral and non-viral transfection of the cell. (**A**) Lentiviral and adenoviral transfection. These are two common viruses used for gene therapy transfections. The packaged nucleic acid escapes from the endosome and is directed to the nucleus if DNA is packaged or is replicated into DNA if RNA is packaged and directed into the nucleus, where integration into the genome occurs. (**B**) Non-viral vector transfection. Here, the packaged DNA is once again released from the endosome; DNA is directed into the nucleus; mRNA remains in the cytoplasm. Small RNA molecules, such as siRNA, will also enter the nucleus [5].

**Figure 2 pharmaceutics-14-00165-f002:**
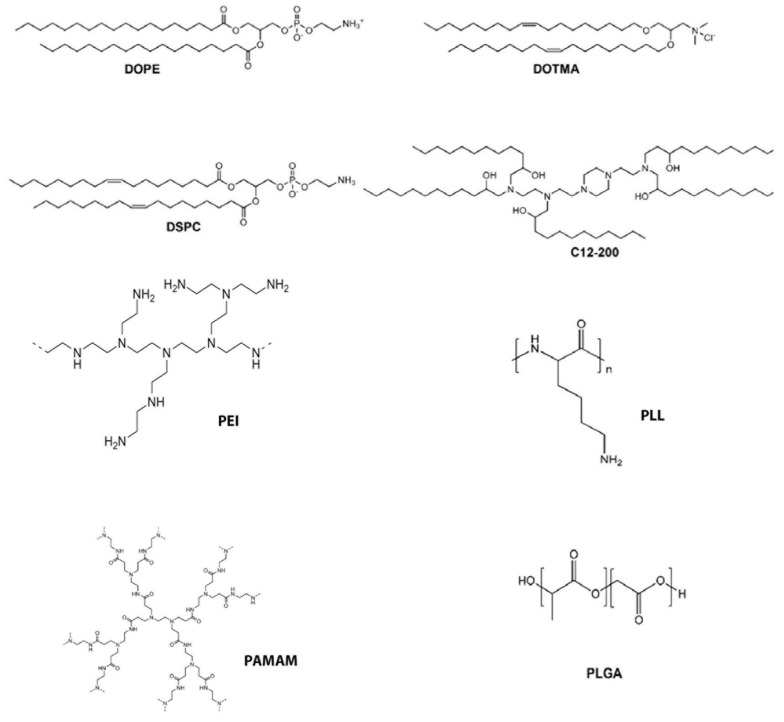
Chemical structures of nucleic acid delivery vehicles described in Table 1.

**Figure 3 pharmaceutics-14-00165-f003:**
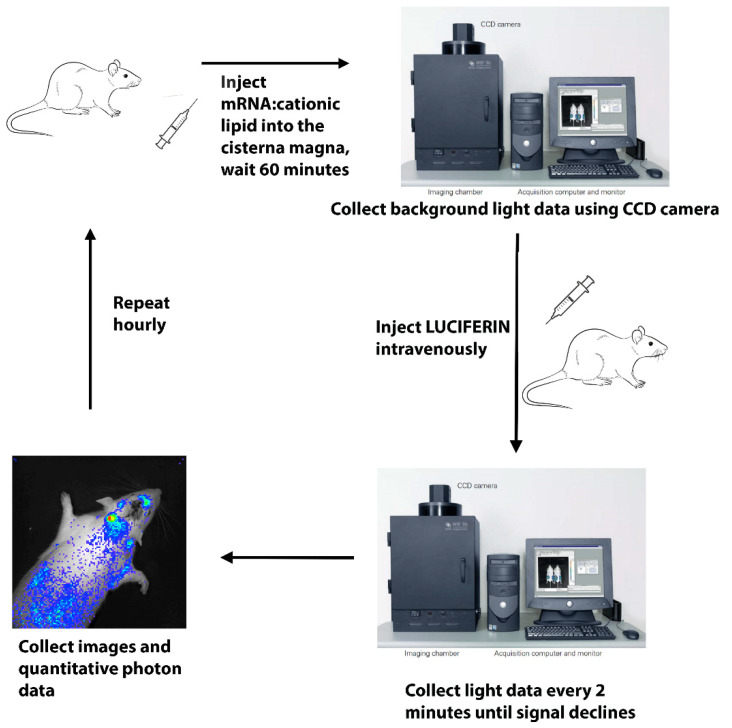
Flow diagram of the in vivo imaging experiment. A luciferase mRNA;cationic lipid complex was slowly injected into the cisterna magna of anesthetized rats. After 1 h, rats were placed into a light tight box and their background light was measured. The animals were then intravenously injected with luciferin and quickly returned to the light tight box. Emitted light was again quantified, and photos were taken of the rats with superimposed images of the emitted light. Emitted light was measured every 2 min for 1 min until only background levels of light were detected. This short time course was repeated hourly until only background light was detected, sometime around 4–7 h after the initial lipid complex injection into the CSF.

**Figure 4 pharmaceutics-14-00165-f004:**
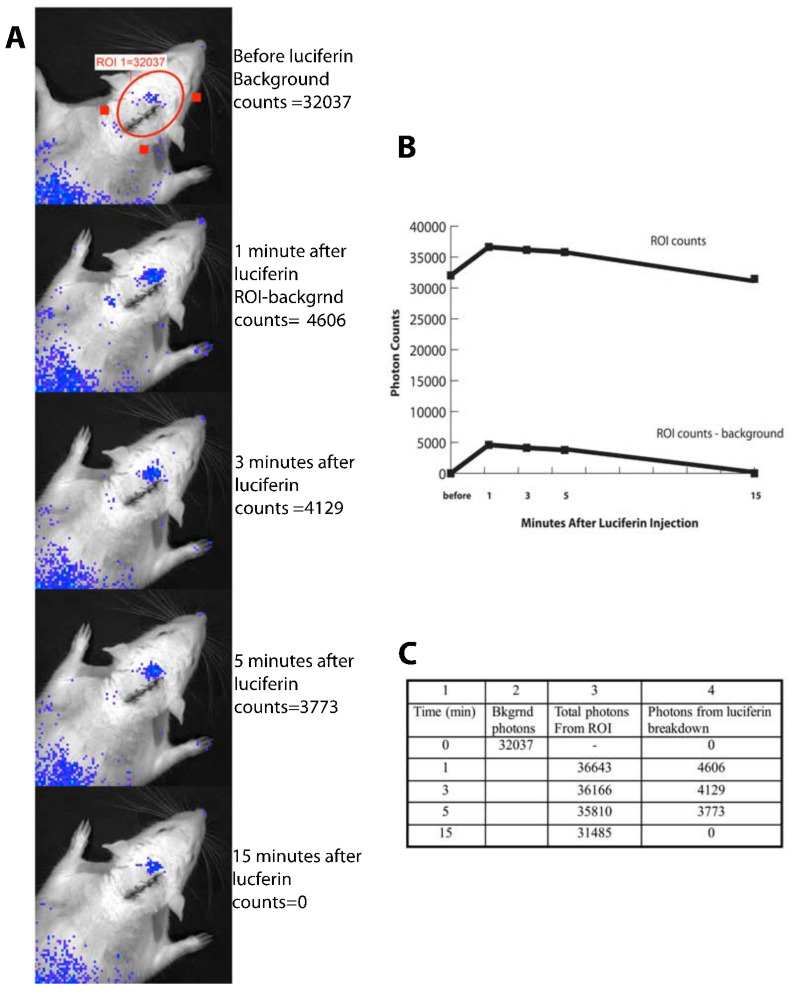
Time course demonstrating the duration of luciferase activity after a single intravenous injection of its substrate, luciferin. Frames (**A**–**C**) are three different representations of the time course data. (**A**): Images from the IVIS living image software of light emission from the head of the rat. (**B**): Emitted photon counts shown in graph form. (**C**): Photon counts of emitted light collected by the cooled CCD camera of the IVIS imaging system. Background photon counts were collected before luciferin injection and subtracted from counts after luciferin dosing.

**Figure 5 pharmaceutics-14-00165-f005:**
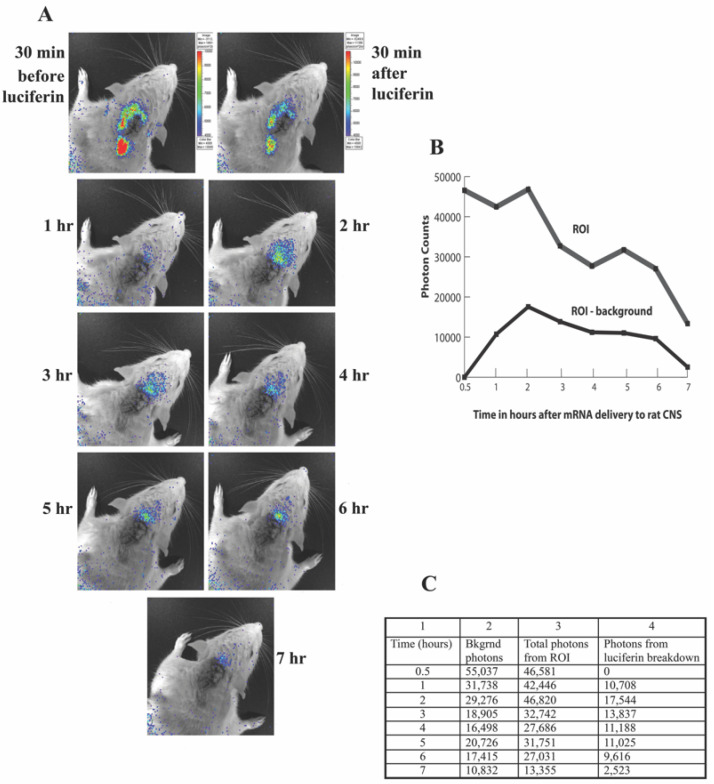
Hourly time course after transfection with mRNA-fLUC. Starting 1 h after transfection of the liposome into the CSF of a rat, the animal was given an intravenous dose of luciferin and imaged every 2 min until light was no longer detected. Peak counts of photon emissions from each hourly time point are shown in the figure (after subtraction of background photon counts). Frames (**A**–**C**) show the data from the time course in three representations. (**A**): IVIS images showing light emission over the rat head. (**B**): Graphic representation of the counted photon emissions from each hourly time point. (**C**): Table showing the hourly photon emission counts.

**Figure 6 pharmaceutics-14-00165-f006:**
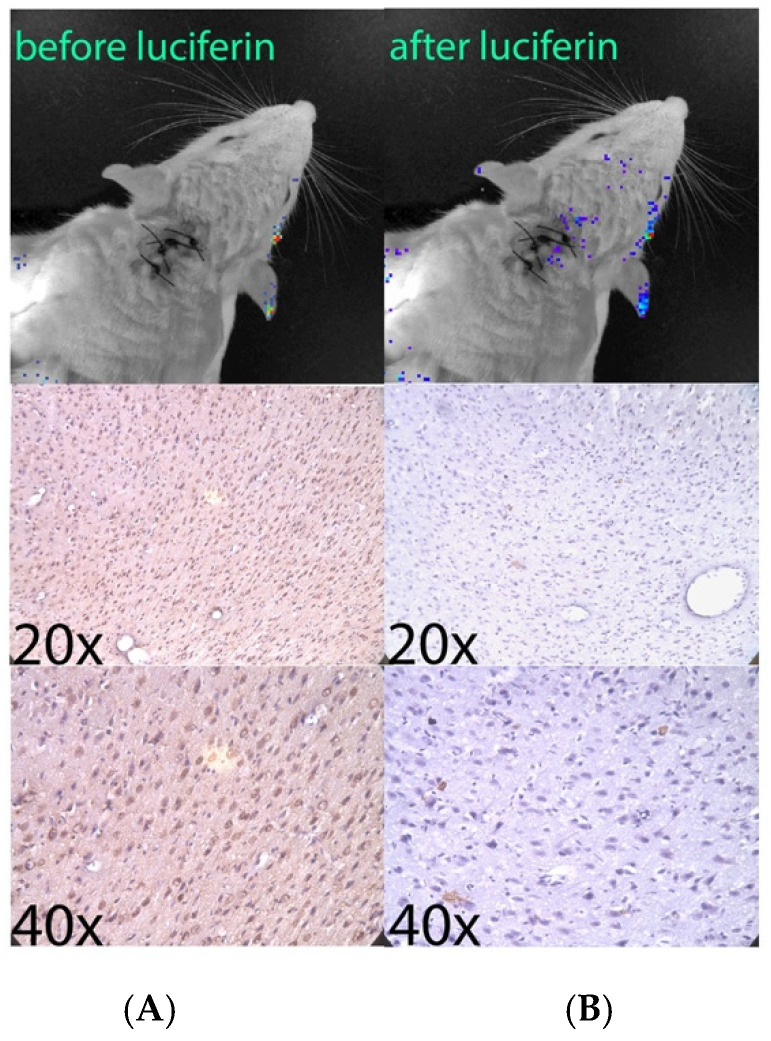
Immunohistochemistry showing the widespread presence of luciferase expression in the brain of a rat after a single cisterna magna injection of MLRI:mRNA-fLUC, as well as a control, non-transfected rat. Rats were infused with 8 μg mRNA that encoded for luciferase in a 120 μL formulation with cationic lipid, using a syringe pump over 30 min. The animals were euthanized 3 h after in vivo imaging and were perfused, sectioned (30 μL), and stained using a monoclonal antibody and immunofluorescent secondary Ab. Left panels: (**A**): IVIS image of the transfected animal prior to intravenous luciferin injection. 20× and 40× control sections are below (**A**). Right panels: (**B**): IVIS imaging of the transfected animal after intravenous luciferin dosing. 20× and 40× sections from the transfected animals are below frame (**B**). Dark staining representing expressed luciferase is seen in the sections from the transfected animal, while there is no such staining in the sections from the control animal.

**Table 1 pharmaceutics-14-00165-t001:** Properties of effective nucleic acid delivery vehicles. This table shows necessary properties of effective cationic lipid- and polymer-based nucleic acid delivery vehicles where newer materials have been successful. Chemical structures are shown in Figure 2 below.

Property	Materials Examples
Packaging of DNA or RNA	Liposomes Cationic lipids e.g., DOTMA, DOPE, DSPC [24], C12-200 [25] Polymers e.g., PEI, PLL, PAMAM, PLGA [24] Organic molecules e.g., chitosan [31], hyaluronic acid, cholesterol, extracellular vesicles [32] [Obrien, 2020]
Stability	Cholesterol, PEG, albumin
Endosomal Escape	Melittin pH sensitive materials PEI, PEG Amines on lipids and polymers

**Table 2 pharmaceutics-14-00165-t002:** Optimization of transfection and light detection: Light detection using IVIS imaging required optimization of the signal from transfected rats as well as background light emission. The table below summarizes the steps taken to optimize our luciferase specific light signal.

Problem	Action	Solution
Biologic activity of mRNA	(1) Improve % of IVT mRNA that can be translated (IVT typically 50%) (2) Screen mRNA for biologic activity before transfection	(1) Add ARCA cap to IVT to produced 80% active mRNA (2) Use only mRNA with high activity demonstrated by cell culture transfection and in vitro luciferase assay
Noise from background light	(1) Rats show varying levels of background light: time in animal housing and fur contribute to noise	(1) Screen all rats prior to transfection by IVIS for low background light emission (2) Shave fur from head

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
