# Peer review of "Non-Viral Delivery of RNA Gene Therapy to the Central Nervous System"

_pharmaceutics, 2022, doi:10.3390/pharmaceutics14010165_

Round 1

Reviewer 1 Report

Hauck and Hecker concisely summarized some critical technical points for nonviral/lipid-based RNA delivery approaches for the CNS. The author supplements the mini-review with useful in-house data to highlight the challenges and potential solutions for pre-clinical experiments. The manuscript is generally easy to read. Some minor comments with hope to aid clarity and readability for broader readerships are listed below:

  • It would generally help the reader if there were complementary illustrations for sections 1-5. This is most needed for readers that are not from the same field. Similarly, having chemical structures for the materials included in Table 1 will also help.
  • Following point 1, the manuscript title suggests the review was about lipids, yet the materials summarized in Table 1, e.g., PEI, PLL, PAMAM, are not lipids. Therefore, the author should reconsider what should be reviewed and the focus of their review article.
  • Figures 1-3 are useful and easy to understand for those familiar with the overall techniques (e.g., IVIS). However, naïve readers who were not involved in similar works will have difficulty understanding it. An illustrated general experimental workflow will be needed to explain the observed data better.

Author Response

We appreciate the comments and suggestions from reviewer 1.  We have added a figure illustrating the cellular pathways of viral and non-viral vector delivery into cells, to assist novice readers.  In addition, all materials listed in Table 1 have structures shown in another added figure, figure 2.

Table 1 does indeed contain materials that are not lipids, and we have expanded Table 1 to include lipids as well as naturally derived or purified materials such as hyaluronic acid and extracellular vesicles.  We have expanded our discussion to include cationic polymers, added some references and a brief comparison of advantages and disadvantages of the most used non-viral vector classes, cationic lipids and cationic polymers.

We have changed the title of the article to “Non-Viral Delivery of RNA to the Central Nervous System.

Finally, again for those not familiar with in vivo imaging experiments, we have added a figure to illustrate the flow of the transfection and time course experiments from our laboratory. 

Reviewer 2 Report

The development of safe and efficient carriers is crucial for gene therapy.  Despite the prevalence and ubiquity of viral vectors usage, this tactic has a few unsettling flaws such as immunogenicity and possible cytotoxicity. The non-viral vectors seem to be the perfect alternative for therapeutic gene delivery. The review of Ellen Hauck and James Hecker summarizes the studies devoted to lipid nanoparticles as a delivery system and gave an overview of the modifications for CNS targeting, providing an interesting and detailed review article. However, some points should be added to improve the manuscript.

MAJOR

1)         Table 1 (page 4, lines 132-133) is representing the data from mentioned reference (26 - Tan et al., doi:10.3389/fnmol.2016.00108). However, the same table is shown in Tan et al., moreover, the origin is a review article of 2016. I wonder whether there are any new approaches (material examples) to tackle the problems of efficient nucleic acid delivery by non-viral carriers? It is recommended to update the table.

2)         The authors focus on studies devoted to lipid nanoparticles, however, information no comparison with other types of non-viral vectors is present. It is recommended to add advantages of lipid vectors over other non-viral vectors.

MINOR

1)         line 11  “Delivery of DNA or mRNA is usually more rapid than viral-mediated delivery...” Did the authors mean «Lipid-mediated delivery»? Cause without this context phrase seems to lose its sense.

2)         line 42  “For clinical applications in which only short-term gene expression is required…”. It would be better for the review to add examples.

3)         line 107  “This has been repeated several times already”. It's not entirely obvious which phrase this refers to. Rephrase, please.

4)         lines 155 “Many RNAs of smaller size (siRNA, saRNA, shRNA,miRNA) are easier to protect and package and possibly easier to achieve efficient transfection. RNA is best for short-term therapy, does not enter the nucleus, and cannot integrate”.  saRNAs target gene promoters, so they need to enter the nucleus. As well as some other RNAs, such as RNA ASO and SSO for splicing correction.

5)         In line 261 authors used “motor” instead of “muscular” mentioning “spinal muscular atrophy”, which seems to be a misprint that should be corrected.

6)         The sentence on lines 301-303 contains “in our laboratory” twice.

7)         line 468. The reference [19] is not full; the needed information should be added. No information was found for “Loss of eyesight in patient after viral vector”.

8)         There are some different term spellings in the text. For instance, “non-viral” is represented in two forms as “non-viral” and “nonviral”. This observation is also valid for “ex vivo”. The terms “in vitro”, “in vivo”, and “ex vivo” should have the same formatting.

Author Response

We appreciate the comments and suggestions from reviewer 2.  We have expanded Table 1 to add cationic polymers and some of the naturally derived materials such as extracellular vesicles now being complexed with nucleic acids as non-viral vectors.  We have added a figure containing structures of the examples listed in the table and made a brief comparison of advantages and disadvantages of cationic lipids (broadly including LNP) and cationic polymers, the two most used types of non-viral vectors.

Line 11. The sentence has been corrected to include “lipid-mediated”

Line 42. Some examples of clinical uses for short-term gene expression have been added.

Line 107.  The sentence “This has been repeated many times” has been deleted as it did nor make sense.

Line 155. The reviewer is of course correct that some RNAs must enter the nucleus to function, and the discussion has been corrected to reflect that mRNA does not need to enter the nucleus, but other RNA structures must be delivered to the nucleus.

Line 261. The error or “motor’ for “muscular” has been corrected.

Line 301-303.  The sentence has been corrected to eliminate one “in our laboratory”

Line 468. The reference for the article describing a patient’s loss of eyesight is now complete.

Throughout the manuscript, corrections have been made to use only the form non-viral. In addition, in vivo, in vitro and ex vivo are also consistent throughout.

Reviewer 3 Report

This manuscript has an important clinical message and should be of great interest to the readers. It presents the main findings and conclusions, but it is not clear whether the authors are showing experimental results or a bibliographic review. If the authors aim to show an experimental result, they should enlarge the studies. This manuscript is not ready for publication in Pharmaceutics. Nevertheless, there are some comments that the authors need to address.

  1. The literature review should be reviewed.
  2. Lines 105-107: “Nonviral vectors have the benefit of lower immunogenicity, and virtually no limit to the size of the delivered nucleic acid. This has been repeated several times already.” These assertions are not entirely true. Please justify these statements and include references in the text that support them.
  3. Line 107: “This has been repeated several times already.” This phrase is out of context.
  4. Lines 158-159: “Increased transfection efficiency in non-dividing cells such as neurons makes RNA the preferred nucleic acid for transfection of the CNS”. Authors should insert references to support this assertion.
  5. Lines 232-233: “The alternate approach is to adapt the drug to use known BBB transport systems. Both methods have experienced successes.” Authors should insert references to support this assertion.
  6. Lines 239-240: “Pfeifer et al were able to inject RNAi designed to suppress prion protein expression in scrapie-infected mice.” Authors should insert reference about the study of Pfeifer et al.
  7. Insert references 46 and 48 before a period.
  8. Lines 282-283:” Nasal delivery of the therapeutics has been demonstrated in mouse models, and likely works by transient disruption of the BBB at high dose”. Authors should insert references to support this assertion.

Author Response

We appreciate the comments and suggestions of reviewer 3.  The manuscript is presented as a mini-review of non-viral vectors delivery to the CNS.  The data presented from our laboratory is meant to add to this review by highlighting some of the challenges (background noise and confirmation of vector localization) and possible solutions to pre-clinical work delivering gene therapy to the central nervous system.

We have added 23 recently published references to the review to add to the completeness of the literature review.

Lines 105-107.  We agree with the reviewer and as these remarks did not add to the paragraph, we deleted the sentences from the paragraph.

Lines 158-159. A reference has been added to support the assertion that RNA is more efficiently transfected into non-dividing cells than is DNA.

Lines 232-233.  References have been added that demonstrate the use of known BBB transport systems.

Lines 239-240. Pfeifer et al reference has been added to the discussion of the work presented therein.

References have all been placed before sentence periods.

Lines 282-283.  References have been added which demonstrate the method of high dose intranasal delivery of non-viral vectors in mouse models to pass the blood brain barrier.

Round 2

Reviewer 2 Report

The authors significantly improved the manuscript. Now it is ready for publication.

Author Response

Thank you.  We appreciate your constructive comments

Reviewer 3 Report

I consider that the experimental work should be independent of the literature review. Therefore, it would be better to analyze both works in depth and present them separately, so I still do not see the adequacy of this manuscript with the two types of publications that are not fully complete.

Author Response

We appreciate your feedback.  We disagree with your assessment that the review should not include unpublished data, particularly data that directly relates to the topic. In addition, this was an invited review manuscript and our planned inclusion of this data was discussed with the editor at the time of acceptance of this invitation.  Finally, reviewers 1 and 2 did not object to the presentation of our data.  In fact, reviewer 1 felt the inclusion of this data positively added a dimension to the article.  We will present our point of view to the editor, and expect that at this point a decision about publication will have to come from this level.